# Superior Damage Tolerance of Fish Skins

**DOI:** 10.3390/ma16030953

**Published:** 2023-01-19

**Authors:** Emily Zhang, Chi-Huan Tung, Luyi Feng, Yu Ren Zhou

**Affiliations:** 1State College Area High School, State College, PA 16801, USA; 2Department of Materials Science and Engineering, Massachusetts Institute of Technology, Cambridge, MA 02139, USA; 3Department of Engineering Science and Mechanics, The Pennsylvania State University, State College, PA 16802, USA

**Keywords:** strain stiffening, collagen, skin, damage-mechanics, biomechanics

## Abstract

Skin is the largest organ of many animals. Its protective function against hostile environments and predatorial attack makes high mechanical strength a vital characteristic. Here, we measured the mechanical properties of bass fish skins and found that fish skins are highly ductile with a rupture strain of up to 30–40% and a rupture strength of 10–15 MPa. The fish skins exhibit a strain-stiffening behavior. Stretching can effectively eliminate the stress concentrations near the pre-existing holes and edge notches, suggesting that the skins are highly damage tolerant. Our measurement determined a flaw-insensitivity length that exceeds those of most engineering materials. The strain-stiffening and damage tolerance of fish skins are explained by an agent-based model of a collagen network in which the load-bearing collagen microfibers assembled from nanofibrils undergo straightening and reorientation upon stretching. Our study inspires the development of artificial skins that are thin, flexible, but highly fracture-resistant and widely applicable in soft robots.

## 1. Introduction

Skin is a multifunctional organ that provides protection to animals from their living environment by regulating body temperature and sensing external stimuli [1]. Skin is made of three layers: epidermis, the top layer; dermis, the middle layer; and hypodermis, the bottom layer [1]. The epidermis is the water-resistant outer layer of skin and the body’s first line of defense against environmental attacks, ultraviolet radiation, bacteria, and other pathogens. It is also responsible for cell renewal. The dermis is responsible for the structure and mechanical properties of skin and mostly composed of the proteins collagen and elastin. Collagen is the major load-bearing structure [2,3], while elastin provides flexibility for the skin [4]. The lower layer of the dermis, the stratum compactum, consists of an orthogonal cross-ply arrangement of collagen microfibers with well-defined angles (~40–60°) relative to the long axis (length) of the fish [5]. For scaled skins, each scale is typically embedded in the dermis and projected out through the epidermis and contributes to the puncture resistance of the skin [6,7]. The hypodermis is the layer of skin where fat is deposited and stored. The presence of multiple layers allows the skin to achieve many functions.

The robust mechanical properties of skin are essential to ensure its function as a self-healing protective layer after being subject to mechanical loads such as abrasion and tearing. The skins of different animals have been studied [8,9,10,11], which are often considered to be nonlinearly elastic. Far less is known about the plasticity and damage tolerance of skin. Because skin is a polymeric structure with a hierarchical order [4,10], plastic deformation likely originates from the rearrangement of the collagen microfiber network. Indeed, Diamant et al. [12] and Fratzl et al. [13] noted the uncrimping of rats’ tail collagen microfibers with increasing tensile loading at small tensile strains of around 0–3%. How collagen microfibers in skin rearrange at high strains of tens or hundreds of percent, and their effect on mechanical properties at these strains, however, has been previously unexplored.

In this study, we characterize the mechanical properties of fish skins under tensile loading.

Fish skin must endure the harsh aquatic environment in very dynamic motions and processing damage tolerant skins is essential for survival. How collagen microfibers in skin rearrange at high strains, and these rearrangement leads to superior mechanical properties and damage tolerance, have been previously unexplored. It is well characterized that the mechanical properties of fish skins are location dependent, as experimentally observed for tensile stress–strain curves of striped bass skin [5] and Chinese sturgeon skin [14]. However, our study here focuses on the damage tolerance mechanism of the skin in general, not on specific locations. We hypothesize that though the mechanical properties are location dependent, the deformation mechanism and damage tolerance strategy are location invariant. Our measurements show that fish skin is highly ductile, nonlinear, and exhibits a strain-stiffening behavior. Loading and unloading cycles show that fish skin undergoes large plastic deformation with significant hysteresis. Stretching can hardly lead to the expansion of pre-cut holes and edge notches on the fish skin; instead, stretching effectively eliminates the stress concentration near the defects, demonstrating that the skin is highly damage tolerant. To explore the microscale mechanism behind the macroscale mechanical properties, an agent-based model was developed. The out-of-plane crimping and the in-plane reorientation of collagen microfibers are widely regarded as the origin of the nonlinear elastic response of collagen network [3,10,12,13,15]. Although molecular dynamics simulations provide the mechanical response of straightening for a single collagen fibril [16], the tissue scale microfiber movement under tensile test has not been investigated. Considering that the collagen network is the major load-bearing structure in the skin, we develop an agent-based model of the collagen network. Our agent-based simulations show that the superior mechanical properties of fish skins can be attributed to the rearrangement of the collagen microfibers network upon stretching.

## 2. Experimental Investigation

### 2.1. Experimental Procedures

Our apparatus setup included a MARK-10 loading apparatus and force gauge, optical microscope, and specimen samples. Fresh striped bass fish were acquired from a local fish store (Cambridge, MA, USA). The fish were about ~20 cm in length and ~8 cm in width and kept hydrated on ice before sample collection. Sample collection was conducted within a few hours of purchasing. Two rectangular tissue samples of ~5–7 cm length and ~2–3 cm width running from anterior to posterior of each fish was then extracted, one sample from each side of a fish. Flesh attached to the bottom surface of each rectangular tissue sample was carefully removed. Fish skins were removed from fish bodies within 2 h of purchase, and stored in water around 0 °C before being shaped into tensile test specimens. The thickness of the obtained skin sample was ~0.3 mm measured using a caliper with 0.01 mm resolution. Strips of 1 cm width and ~5–7 cm length were then cut parallel to each other from this skin sample. These 1 cm-wide strips were tensile test samples, with the loading axis parallel to the length. The in-plane collagen microfiber orientations relative to the tensile axis were the same (~40–60°) for all the samples [5]. In total, three fish of similar size were used, and each fish provided six 1 cm-wide test samples. The samples were kept hydrated with fresh water between sample collection and tensile testing.

During tensile testing, the strips were clamped to the testing apparatus with a grip at each end, and uniaxial tension was applied along the length direction until the strips ruptured. All tensile tests occurred at room temperature of around 20 °C. For each specimen, the stretch rate was kept at 10 mm/min. The stress is calculated by σ=F/(wt), where w is the width and t is the thickness. The engineering strain is calculated by $\varepsilon =\Delta l/l_{0}$, where the elongation is $\Delta l=l-l_{0}$, and l and $l_{0}$ are the stretched and original lengths of the specimen, respectively. The stretched length was calculated by multiplying the grip–grip displacement rate by the elapsed time. The zero-strain grip-grip displacement occurred when tensile load first started to increase from zero. As the strain calculation was based on the recorded grip–grip displacement, sample–grip slip may cause strain measurement inaccuracy. However, the error due to sample–grip slip is negligible in our experiments. The fish skin samples in our experiments reach a grip–grip displacement of a few cm before rupture occurs, whereas the sample–grip slip that we observed was at most a few mm. The different orders of magnitude of the displacement and slip gives us great confidence in our strain measurements.

### 2.2. Tensile Response of Fish Skin

We performed several tensile tests with the fish skin samples. A typical stress–strain relation is shown in Figure 1. The stress–strain relation is highly nonlinear. The stiffness of the skin—the slope of the stress–strain curve—increases with strain, indicating a strain-stiffening effect. The infinitesimal Young’s modulus of one fish skin sample is 13.3 MPa at 0% strain, but can reach as high as 40.4 MPa at 20% strain. Another fish skin sample showed similar results, with 13.0 MPa at 0% strain and 32.5 MPa at 20% strain. The fish skin is highly extensible with a rupture strain of more than 40%. The tensile strength of the skin can reach 14 MPa. From our experiments, failure often occurs near the clamped ends of the testing samples, but rarely in the center, possibly due to strain localization at the clamped ends. The fish skin samples were not descaled since the natural tensile response of the skin was of interest in this study. It is known that scales provide out-of-plane puncture resistance of the skin [5]. Scales also influence tensile response significantly. The tensile specimen can be decomposed into scaled regions and non-scaled regions. Due to the much higher stiffness of the scaled regions relative to the non-scaled regions, strain is mainly localized at the non-scaled regions. Therefore, at a given stress, the local strain at non-scaled regions may be considerably higher than the strain measured in this study.

While it has been well characterized that the tensile strength and Young’s modulus of bass skin may vary significantly depending on the sample’s location on the fish [5], our study focuses on the change in mechanical properties and structure with increasing tensile strain, i.e., the deformation mechanism and the damage-tolerance strategy of the skin that are shared by skins at all locations, rather than the differences in mechanical properties and structure between different locations on the fish. Therefore, knowing the mechanical responses between the samples may vary, comparisons of experimental results are constrained to one sample at different tensile strains.

### 2.3. Loading-Unloading Reveals Strain-Dependent Mechanical Properties

In real-life conditions, fish skins undergo frequent mechanical loading and unloading impacts, such as during cyclic motions of the fish’s body. To study fish skins under mechanical cycling, we applied tensile loading and unloading cycles to the fish skin specimens. The specimens were first stretched to a certain force below the failure strength and then unloaded until the applied force returned to zero, then reloaded to a higher peak force until fracture occurred.

To see whether stretching may cause plastic deformation, we applied tensile loading-unloading cycles to two specimens (Figure 2). For the first loading–unloading specimen, we unloaded the specimen to a 21% tensile strain (blue curve). Upon unloading to zero force, the specimen retained a plastic strain of 13.5%, suggesting an elastic strain of only 7.5% during loading. The large amount of plastic strain suggests permanent changes inside the skin. For metals and other crystalline materials, the unloading curve usually has the same slope as the elastic portion of the loading curve. However, for the fish skin, the unloading and reloading stiffness is much larger than the previous loading stiffness, indicating microstructural changes such as collagen microfiber network rearrangement during loading have already taken place in the previous loading process. The area encompassed by the loading and unloading curves is the energy dissipated during the cycle. The significant hysteresis area in the stress–strain curve further indicates strain-dependent mechanical properties and microstructural rearrangement during loading. Reloading closely followed the previous elastic unloading curve until the previous maximal loading point was reached, showing that the mechanical properties do not change significantly during unloading–reloading steps up to the previous maximal point. Finally, continued loading beyond the previous maximal point leads to further plastic deformation.

For the second specimen, we conducted two loading–unloading cycles (orange curve). For this specimen, the initial Young’s modulus was smaller than that of the first specimen, though strain-stiffening was observed for both specimens. Hysteresis occurred during both loading–unloading cycles. In each cycle, appreciable plastic deformation was generated in the tensile specimen. In the first cycle, the plastic strain was 14% with a total strain of about 19%, suggesting an elastic strain of 5% built in the specimen during the first loading step. In the second cycle, the total strain (relative to after the first unloading) was 10% and the plastic strain generated was about 3%, leading to an elastic strain of 7%. Apparently, the plastic strain and hysteresis generated in the second cycle were markedly smaller than those in first cycle, possibly because of the strain-stiffening effect, making it increasingly more difficult to generate plastic deformation even at a higher applied load.

We note that the loading–unloading hysteresis is opposite from the classical Mullins effect observed in rubbers as the reloading stress–strain curve below the maximum stain experienced in its prior strain history does not show softening, but rather stiffening. This indicates that even at small strain, the collagen network undergoes large-scale permanent rearrangement that causes the stiffening.

### 2.4. Slits and Holes Are Insensitive to Stretching

To test the fish skins’ resistance to stress concentration rupture, we introduced holes to the specimens (Figure 3A). The holes were irregular in shape but close to each other (Figure 3(A2)). In general, the presence of a hole causes stress concentration at the periphery of the hole, which then causes rupture of the ligament between the holes. The stress concentration scales with the equation $\alpha =1+\frac{2a}{b}$, where a is the radius of the hole along the width direction and b is the radius of hole in the length direction (stretching direction). For a circular hole, the stress concentration factor is 3. Unexpectedly, we found that stretching did not cause expansion of the holes or the rupture of the ligament between the holes. Instead, upon stretching, the holes were elongated along the stretching direction with increasing b but decreasing a (Figure 3(A2)). This significantly reduced the stress concentration factor α at the holes. In the end, the hole became a slit with a≪b and α≈1, which means a nearly eliminated stress concentration due to the hole geometry (Figure 3(A3)). Continuing stretching did cause rupture near the region where holes were introduced, probably due to reduced cross-section area. However, the failure strengths/strains were not significantly lower than the pristine specimens (Figure 3(A1)).

We also tested whether edge notches could cause significant damage to the skin (Figure 3B). We introduced an edge notch to the specimens. The edge notches were about 5 mm-long, half of the width of the samples. We observed that during stretching the edge notch did not propagate as a crack. Instead, the notch became increasingly blunted and finally eliminated (Figure 3(B2,B3)). Necking then occurred at the blunted region and further stretching caused rupture in this region. There was no extension of the initial notch length. These experiments indicate that fish skins are highly tolerant to defects and mechanically very tough, a property which has previously been demonstrated in other materials such as bone and tough hydrogels [17,18].

To quantify the toughness, tensile tests of the notched and pristine specimens were carried out to determine the fracture energy of the fish skins [19,20]. In each test, two specimens with the same dimensions were prepared and stretched; one was unnotched and the other was pre-notched. We stretched the unnotched specimen to rupture and determined the work of fracture, which is the area under the stress-stretch curve w(λ), where $\lambda =\frac{l}{l_{0}}=1+\varepsilon$ is the mechanical stretch. $\lambda_{f}$ is the stretch at which the unnotched sample ruptures. We then stretched the pre-notched specimen and found $\lambda_{c}$ as the stretch at which the notched sample ruptures. Apparently, $\lambda_{c}<\lambda_{f}$. The fracture energy is then determined by $\Gamma =l_{0}w\left(\lambda_{c}\right)$, where $l_{0}$ is the length of the skin strips in the undeformed state. This formula for the fracture energy assumes that (1) the notch further opens when the stretch at its tip is $\lambda_{c}$, and (2) the elastic energy stored above and below the notch tip is converted to notch opening energy when the tip stretch reaches $\lambda_{c}$. The stored elastic energy per unit volume above and below the tip just before opening is $w\left(\lambda_{c}\right)$, so the notch opening energy (i.e., fracture energy) per unit area is $l_{0}w\left(\lambda_{c}\right)$.

For the first edge-notched specimen of initial length $l_{0}=7$ cm, $\lambda_{c}=1.28$ (Figure 3(B1)), $w\left(\lambda_{c}\right)$ can be found from the strain–strain curve of the unnotched specimen. The fracture energy we measured was Γ=0.0784 MJ/m^2^. For the second edge-notched specimen with $l_{0}=7$ cm, $\lambda_{c}=1.31$ and Γ=0.09765 MJ/m^2^. The measured fracture energy (Γ) and the work of fracture of the unnotched sample ($w_{f}=w\left(\lambda_{f}\right)$) defines a characteristic length scale, $l_{f}=\Gamma /w_{f}$, termed as the flaw-sensitivity length [17] below which the stretchability of the skin is insensitive to the flaw. For a pre-notch of length less than $l_{f}$, the notch has negligible effect on the critical stretch of the sample, and the sample is flaw insensitive; on the other hand, for a precut notch of length longer than $l_{f}$ the notch significantly reduces the critical stretch of the sample. As the strain at failure of both notched and unnotched samples is around 30%, the 5 mm pre-notch length used in this study serves as the directly measured lower bound of flaw-sensitivity length. As the total work of fracture for a similar specimen without an initial cut is $w_{f}=4.30$ MJ/m^3^, the calculated flaw-sensitivity lengths of the first and second edge-notched specimens are 18 mm and 22 mm, respectively. These calculated values are probably an upper bound, since unnotched specimens which do not fail near grips due to stress concentration should exhibit larger $\lambda_{f}$, larger $w_{f}$, and hence lower $l_{f}$. Experiment and calculation rise to a flaw-sensitivity length of $5\;\mathrm{mm}<l_{f}<22\;\mathrm{mm}$, larger than that of most engineering materials [19]. In light of the high flaw insensitivity of the fish skins, we expect that for sufficiently small notches of size less than $l_{f}$, the crack tip will first blunt and the notch will be eliminated upon further stretch, as seen in our experiments.

## 3. Agent-Based Modeling

### 3.1. Construction of an Agent-Based Model

The primary load-bearing structure in fish skin is the collagen microfiber network [3,10], so a model of this network was constructed to better understand the mechanisms behind the experimental observations in the previous sections. In a natural, unstretched state, collagen is arranged to form thick collagen microfibers [21], which adopt an orthogonal cross-ply in-plane arrangement oriented at 40–60° with respect to the fish body length with some degree of waviness in the out-of-plane thickness direction [5] (Figure 4A). While the angles of the collagen microfiber cross-ply may depend on the location on the fish’s body, the elementary microfiber movements are expected to occur at high strains at all regions in the fish skin. As such, our model is based on the in-plane orthogonal cross-ply arrangement and out-of-plane sinusoidal waviness of collagen microfibers in the unstretched state [5], denoted here as a cross-helical structure. Each collagen microfiber was represented by a chain of beads, each bead having a diameter of $r_{0}=1$ μm and a mass of $m_{0}=7.01\times 10^{-16}$ kg. A representative volume element (Figure 4B) was chosen and periodic boundary conditions were applied at the sides of the element. To generate a cross-helical structure, collagen microfibers were initially placed in two orthogonal orientations with a z-direction amplitude and wavelength of 5 and 100 μm, respectively [22,23], and the dimensions of the unit box were 70 μm × 70 μm × 13 μm.

The mechanical properties of collagen chains were determined by the interaction parameters, including intra-microfiber bond potential, intra-microfiber angle potential, and inter-microfiber interaction potential. These parameters were tuned to be consistent with Young’s modulus, persistent length, and adhesion energy, which have been measured experimentally. The intra-microfiber bond potential within was described by a harmonic spring
$$U_{bond}=\frac{1}{2}\;k\;{\left(r-r_{0}\right)}^{2}$$
where r is the distance between a pair of bonded beads, k is the spring constant, and $r_{0}$ is the equilibrium bond length. The spring constant can be expressed by the Young’s modulus E of the collagen microfiber: $k=\frac{A\;E}{r_{0}}$, where A is the cross-section area of the microfibers. In our model, Young’s modulus was given by E=6.5 GPa, based on previous experimental work [24,25]. The angle-bending potential is chosen to match the bending rigidity and persistence length of the collagen microfiber:$$U_{angle}=\frac{l_{p}\;k_{b}\;T}{r_{0}}\left(1+\cos\left(\theta \right)\right)$$
where $l_{p}$ is the persistence length of the microfibers, T=300 K is the temperature and $k_{b}$ is the Boltzmann constant. Based on the published measurements [25], the persistent lengths of collagen chain was set as 10 μm. The interaction between collagen beads from different microfibers was described by the Lennard-Jones (LJ) potential [26], where the energy parameter is set to be ε=11 kcal/mol.

All simulations were performed in LAMMPS. Starting with an initial cross-helical configuration, simulations were performed for 1 ms using NVE ensemble and Langevin thermostat to maintain a system temperature at 300 K until the system reached equilibrium. Then, an additional Berendsen barostat was applied to relax the stress for 100 μs. After the stress relaxation, the pre-equilibrated system was uniaxially stretched by deforming the simulation box along the stretching direction at a strain rate of 10^3^ s^−1^, with NVE ensemble and Langevin thermostat. The stretching direction was initially oriented at 45° with respect to the in-plane directions of the collagen microfibers.

### 3.2. Modeling Results

We used the model to represent the fish skin and to compare its tensile response to the actual skin. The simulated stress–strain curve resembles our experimental data within 20% stretch (Figure 4(C1)). A two-step stretching process was observed in our simulation. At smaller strains, the dominant mechanism was the reduction of out-of-plane waviness and straightening of the collagen microfibers (Figure 4(C2)). Meanwhile, the modulus of tissue scale collagen network increases significantly with the straightening processing. At larger strains when the out-of-plane crimping is depleted, the dominant mechanism is switched to in-plane re-orientation of microfibers along the stretching direction (Figure 4(C3)). With continuing reorientation, the stiffness continues to increase while the hardening effect attenuates compared with the straightening step. As a result of this two-step stretching process, the collagen microfibers became increasingly aligned along the tensile axis, both in-plane and out-of-plane. These aligned microfibers can resist stretching more efficiently than those with the wavy morphology and larger orientation angles with respect to the tensile axis, which explains the strain-stiffening behavior. Straightening and reorientation are possibly both elastic and plastic, since part of the straightening and reorientation process may not be recoverable upon unloading, which contributes to plastic strain during stretching.

It should be noted that the inter-microfiber sliding of the aligned collagen microfibers may be a possible microfiber movement mode at relatively high stretch. Inter-microfiber sliding usually occurs when the microfibers in contact are of different lengths or microfibers are broken during stretching. However, such microscopic structural information is currently not known from experimental imaging. The cross-ply arrangement may impose an interlocking effect that raises the energy required for inter-microfiber sliding. Furthermore, inter-microfiber sliding is of long range since one microfiber may be in contact with multiple microfibers, enabling long-range energy dissipation. Due to the use of periodic boundary conditions large-scale inter-microfiber sliding is not simulated in our model. Nevertheless, the good agreement in the tensile response between our agent-based model and the experimental data indicates that microfiber straightening and reorientation are the dominant modes in the range of the applied strain.

At the notch tip, all three elementary movements of collagen microfibers may be activated at small global strain because of the high stress concentration. Thus, the notch tip becomes more stiffened than other regions under the applied stretch, effectively avoiding further localized deformation and stress concentration at the notch tip. The local stiffening mechanism, combined with the long-range energy dissipation of the collagen microfibers, explains the superior damage tolerance of fish skin.

## 4. Conclusions

In summary, the mechanical properties of fish skins were characterized by testing skin strips in tension. We found that the fish skins become stiffer with stretching. More importantly, the fish skins are highly damage tolerant. This was demonstrated by the phenomena that stretching can eliminate the stress concentration near the holes and edge notches but cannot propagate or expand these defects. In particular, the flaw-sensitivity length for the fish skin was calculated to be in the order of 10 mm, superior to most engineering materials. We attribute strain stiffening and the superior damage tolerance to the elementary collagen network movements, namely, microfiber straightening and reorientation, as verified by our agent-based modeling. The fundamental understanding of our study also provide guidance to the synthesis of multifunctional artificial skins [27] for soft robots that are mechanically robust and damage tolerant.

## Figures and Tables

**Figure 1 materials-16-00953-f001:**
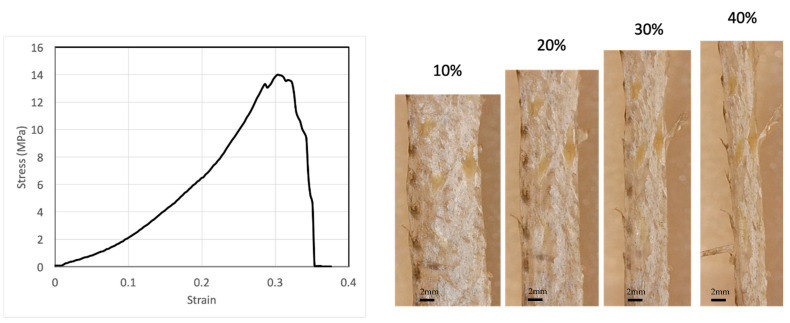
A typical stress–strain relation of fish skins together with microscope images of the skin at selected tensile strains, showing that fish skin is highly extensible and strain stiffening.

**Figure 2 materials-16-00953-f002:**
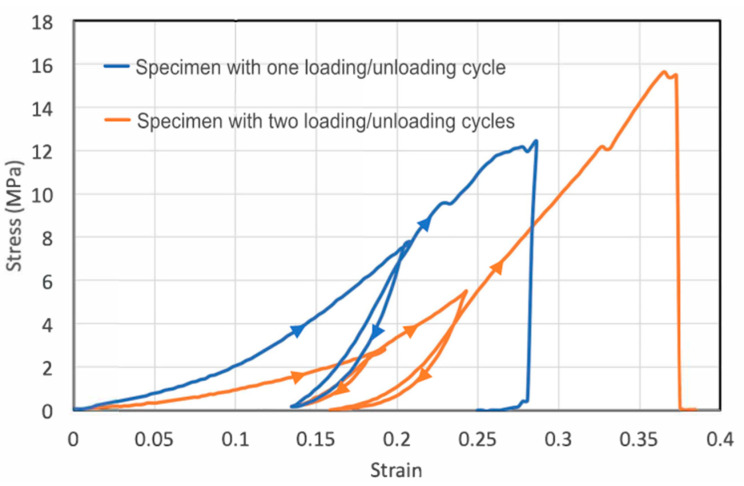
Cyclic loading of the skin strips showing the large plastic deformation and significant hysteresis. The first specimen (blue) was subject to one loading–unloading cycle before reloaded to failure, whereas the second specimen (orange) was subject to two loading–unloading cycles. The loading–unloading directions are indicated by triangular arrows.

**Figure 3 materials-16-00953-f003:**
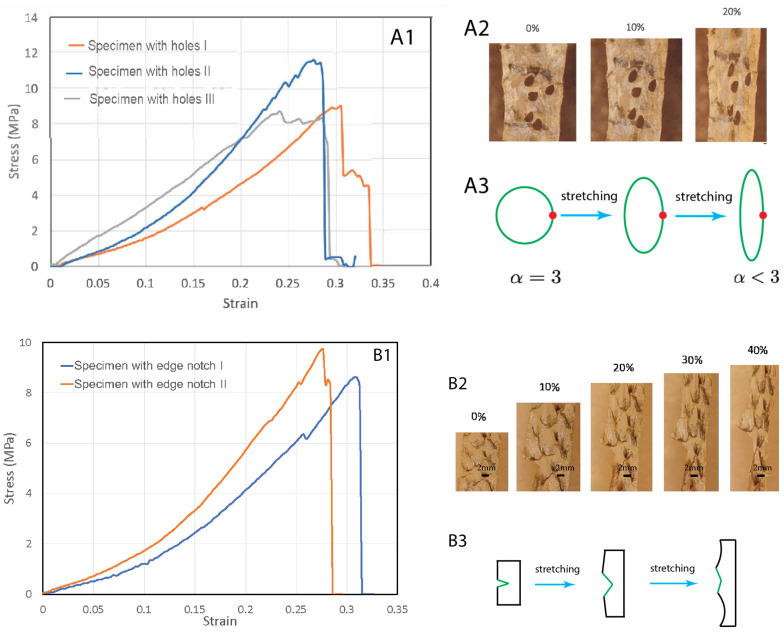
Tear tests of fish skin strips with holes (**A1**–**A3**) and an edge notch (**B1**–**B3**). (**A1**) Stress–strain curves of three specimens with pre-introduced holes. (**A2**) The holes become elongated along the stretching direction, with a reduced stress concentration factor. (**A3**) Schematics of the hole shape change due to stretching, resulting in reduced stress concentration factor. (**B1**) Stress–strain curves of three specimens with a pre-introduced edge notch. (**B2**) Under increasing stretch, the edge notch did not expand, but blunted. (**B3**) A schematic illustration of the edge notch blunting due to stretching.

**Figure 4 materials-16-00953-f004:**
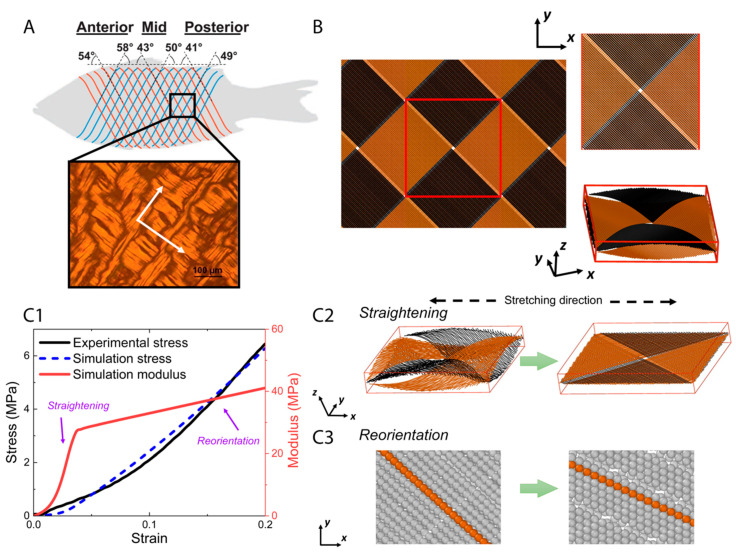
The orthogonal-cross-ply structure of collagen microfibers in fish skin (Reproduced with permission from reference [5]; Copyright 2017 Elsevier) where white arrows indicate the orientations of microfibers in the structure (**A**), geometry setting of agent-based model (**B**), and tensile test results in simulation (**C1**–**C3**). (**A**) The periodic microfibers have two orthogonal principal directions. A corresponding representative volume element was generated (**B**), and then tensile test was applied. The stress–strain curve and modulus as a function of strain, from experimental and simulation (**C1**), show a two-step stretching process. Two primary movements, straightening (**C2**) and reorientation (**C3**) of the microfibers, dominate the tensile response successively, which together elucidate the mechanism underlying strain-stiffening behavior of fish skin.

## Data Availability

Data is available upon request from corresponding author.

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
