# Peer review of "Superior Damage Tolerance of Fish Skins"

_materials, 2023, doi:10.3390/ma16030953_

Round 1
Reviewer 1 Report
The authors need to discuss their results in light of the papers previously published on fish skin mechanics and the literature on mechanical behavior of human skin and tendons. The discussion of these topics are needed to properly interpret their results.

Reviewer 2 Report
This paper describes experimental work to determine the mechanical properties of bass fish skins and theoretical work to understand structural changes behind the nonlinear mechanical behavior of the skin under tensile loading. The topic of the study is interesting and the findings obtained would be of interest to those who study soft tissue biomechanics as well as materials scientists and engineers. However, there are a number of concerns that should be addressed before further consideration for publication in the journal.
1. The fish were purchased from a local fish store and used within a few hours of purchasing. How fresh were these fishes? How long were these fish placed in the store before the purchase? Any history of freezing and defrosting? Such information should be presented to justify your sample preparations and experimental results.
2. How were the width and thickness of the samples measured to calculate nominal stress?
3. How was the zero strain position defined?
4. Testing temperature?
5. Did you apply preconditioning to the specimens before stretching to failure? If not, please explain your reasons.
6. Why did the authors not prepare samples in a dumbbell shape, which is a very common technique in the mechanical testing of soft tissues and materials?
7. Quantitative data should be presented in mean±SD with the number of specimens.
8. The samples mainly failed near clamped ends but rarely in the center. This indicates a great amount of force was applied by grips, and thus the deformation should be greater near the clamped ends than at the center. This potentially complicated results in two points. First, the failure load measured and reported here is not equal to the true failure load. The true failure load and thus true tensile strength should be higher than that reported. Second, the sample deformation should have not been homogeneous along the length of the specimens. Therefore, the strain values determined were a mixture of tissue deformation by tensile loading and stress concentration at the clamps, resulting in an overestimation of strain values. The strain should be determined using strain markers placed on the tissue surface. Overall, the quality of mechanical characterization was not sufficiently high. The authors should conduct tensile tests by addressing these issues or should provide reasons to justify their methodologies.
9. How many samples were used in loading-unloading experiments? Was the reproducibility of the results confirmed with multiple samples?
10. Line 146 “ligament”?
11. Line 165 Could you give some examples of “other materials”?
12. Line 176 Why can w(lc) be found from the stress-strain curve of the unnotched specimen but not that of the notched specimen?
13. Lines 211-212 The information taken from references 15 and 16 was from human tissues. How fishes and humans can be related in terms of collagen structure?
14. Lines 243-244 “the dominant … straightening of the collagen microfibers” This mechanical behavior seems identical to tendons and ligaments where crimped collagen fibers were firstly straightened at a small strain when they are stretched from an unloaded state. The authors could expand this section by discussing the similarity of mechanical behaviors between fish skins and load-bearing collagenous tissues in mammals.
15. Figure 4A This subfigure is almost a direct copy of Figures 3b and 3c in reference 5. The authors may need approval from Elsevier to use them. Otherwise, you draw your own schematic illustration.
Reviewer 3 Report
The authors investigate the mechanical behavior of striped bass fish skins. Undamaged and damaged (with holes and notch) samples of skins were tested under tensile. An agent-based model of collagen network was also introduced. I would like to raise some issues for further clarification as summarized below:
1) The originality of the scientific contribution should be clearly stated in the manuscript. I cannot find the novelty of the manuscript. The authors are encouraged to improve the introduction, showing the novelty of the work.
2) Mechanical properties of biological tissues depend on the time post-mortem and storage temperature. Please, state the number of hours/days post-mortem for your experimental samples.
3) The Authors wrote: “We performed several tensile testes…” How many samples of fish skins were tested?
4) Figure 2 shows a Mulling effect in the uniaxial cyclic extension tests. The stress softening can be associated with damage. This information should be included.
5) Irregular holes were introduced in skin samples. Why? How was the procedure performed?
6) It is not clear how the strain-stiffening and damage tolerance of fish skins can be explained by the developed agent-based model. It should be explained. Section 3 must be improved. All parameters of U_angle must be defined. Epsilon is denoted by strain and energy parameter.
7) A comparison between experimental and simulation results is shown in Fig. 4. The input data of simulation are required.
Round 2
Reviewer 1 Report
see attached

Reviewer 2 Report
1. No pre-freezing or defrosting process was involved. Then fish skins were isolated from the fish within 2 hours of the purchase and stored in natural cold water (close 0 degree) before they were taken out for tensile experiments.
This should be in Experimental procedures.
2. We used caliper to measure the thickness, with resolution of 0.01 mm.
This should be in Experimental procedures.
3. The zero-strain position was determined at the point when the load started to increase from zero.
This should be in Experimental procedures.
4. We tested the specimen at room temperature
This should be in Experimental procedures.
5. Preconditioning is a mechanical treatment in which a small magnitude of cyclic load is applied to specimens before stretching to failure. This is a common procedure in mechanical tests, especially tests for soft biological tissues. Because once the specimens are detached from the body, it is released from the mechanical load, resulting in changes in tissue structure (tissue adapts to stress-free condition). Therefore, before the specimen is tested until failure to determine mechanical properties, it should be subjected to mechanical loading to set the specimen in the environment close to in vivo mechanical environment.
6. In our rectangular sample tests (fig. 1), the fish skin strips underwent quite uniform strain until very close to the rupture point. Therefore, there was no need for us to use dumbbell samples.
Need to show evidence, preferentially images during testing.
7. If we present mean standard deviation of our data, it will show up the location variance, which is not the purpose of our study.
Showing the data in standard mean±SD format does not interfere with the purpose of the study. Please included mean±SD data in the manuscript.
8-1. the failure load measurement is accurate.
The samples failed due to stress concentration at the grips, while the mid-section was not failed. Therefore, a higher load is expected if the specimen fails at the middle part with no stress concentration at the grips.
8-2. observation of the in-situ microscope images collected during tensile testing revealed that, until shortly before rupture, the tensile strain was quite uniform across the sample, and the tensile strain measured by grip displacement is close to the tensile strain deduced from sample geometry changes in the microscope images.
Again, the authors need to show evidence.
9. Two samples, shown in Figure 2, were used in the cyclic loading-unloading experiments.
Indicate this in the main body.
11. bone and tough hydrogels
Should be included in the main body.
12. The model for calculating fracture energy makes the following assumptions:
I can understand your assumptions. It should be included in the main body.
Reviewer 3 Report
Some required changes have been made. In my opinion, the revised manuscript is acceptable for publication.
Author Response
We thank reviewer 3 for his/her time and effort